# Identification of Subclinical Lung Involvement in ACPA-Positive Subjects through Functional Assessment and Serum Biomarkers

**DOI:** 10.3390/ijms21145162

**Published:** 2020-07-21

**Authors:** Bruno Lucchino, Marcello Di Paolo, Chiara Gioia, Marta Vomero, Davide Diacinti, Cristina Mollica, Cristiano Alessandri, Daniele Diacinti, Paolo Palange, Manuela Di Franco

**Affiliations:** 1Dipartimento di Scienze Cliniche, Internistiche, Anestesiologiche e Cardiovascolari- Reumatologia, Sapienza University of Rome, 00161 Roma, Lazio, Italy; bruno.lucchino@uniroma1.it (B.L.); chiara.gioia@uniroma1.it (C.G.); marta.vomero@uniroma1.it (M.V.); cristiano.alessandri@uniroma1.it (C.A.); 2Dipartimento di Sanità Pubblica e Malattie Infettive, Sapienza University of Rome, 00161 Roma, Lazio, Italy; marcello.dipaolo@uniroma1.it (M.D.P.); paolo.palange@uniroma1.it (P.P.); 3Dipartimento di Scienze Radiologiche, Oncologia e Anatomia Patologica, Sapienza University of Rome, 00161 Roma, Lazio, Italy; davide.diacinti@uniroma1.it (D.D.); daniele.diacinti@uniroma1.it (D.D.); 4Dipartimento di Metodi e Modelli per l’Economia, il Territorio e la Finanza, Sapienza University of Rome, 00161 Roma, Lazio, Italy; cristina.mollica@uniroma1.it

**Keywords:** rheumatoid arthritis, interstitial lung disease, subclinical involvement, ACPA, pulmonary function testing, CPET, surfactant protein D, DL_CO_, HRCT

## Abstract

Lung involvement is related to the natural history of anti-citrullinated proteins antibodies (ACPA)-positive rheumatoid arthritis (RA), both during the pathogenesis of the disease and as a site of disease-related injury. Increasing evidence suggests that there is a subclinical, early lung involvement during the course of the disease, even before the onset of articular manifestations, which can potentially progress to a symptomatic interstitial lung disease. To date, reliable, non-invasive markers of subclinical lung involvement are still lacking in clinical practice. The aim of this study is to evaluate the diagnostic potential of functional assessment and serum biomarkers in the identification of subclinical lung involvement in ACPA-positive subjects. Fifty ACPA-positive subjects with or without confirmed diagnosis of RA (2010 ARC-EULAR criteria) were consecutively enrolled. Each subject underwent clinical evaluation, pulmonary function testing (PFT) with assessment of diffusion lung capacity for carbon monoxide (DL_CO_), cardiopulmonary exercise testing (CPET), surfactant protein D (SPD) serum levels dosage and high-resolution computed tomography (HRCT) of the chest. The cohort was composed of 21 ACPA-positive subjects without arthritis (ND), 10 early (disease duration < 6 months, treatment-naïve) RA (ERA) and 17 long-standing (disease duration < 36 months, on treatment) RA (LSRA). LSRA patients had a significantly higher frequency of overall HRCT abnormalities compared to the other groups (*p* = 0.001). SPD serum levels were significantly higher in ACPA-positive subjects compared with healthy controls (158.5 ± 132.3 ng/mL vs 61.27 ± 34.11 ng/mL; *p <* 0.0001) and showed an increasing trend from ND subjects to LSRD patients (*p* = 0.004). Patients with HRCT abnormalities showed significantly lower values of DL_CO_ (74.19 ± 13.2% pred. vs 131.7 ± 93% pred.; *p =* 0.009), evidence of ventilatory inefficiency at CPET and significantly higher SPD serum levels compared with subjects with no HRCT abnormalities (213.5 ± 157.2 ng/mL vs 117.7 ± 157.3 ng/mL; *p =* 0.018). Abnormal CPET responses and higher SPD levels were also associated with specific radiological findings. Impaired DL_CO_ and increased SPD serum levels were independently associated with the presence of HRCT abnormalities. Subclinical lung abnormalities occur early in RA-associated autoimmunity. The presence of subclinical HRCT abnormalities is associated with several functional abnormalities and increased SPD serum levels of SPD. Functional evaluation through PFT and CPET, together with SPD assessment, may have a diagnostic potential in ACPA-positive subjects, contributing to the identification of those patients to be referred to HRCT scan.

## 1. Introduction

Rheumatoid arthritis (RA) is a chronic inflammatory disease, characterized by progressive joint damage and systemic extra-articular involvement [1,2]. Although considered primarily as an inflammatory disease of synovium, extra-articular features are not uncommon and currently represent the leading causes of death among patients with RA, with lung involvement second only to cardiovascular morbidity [3].

In genetically predisposed individuals exposed to various environmental factors, the disease develops progressively following a multistep process, starting with the breaking of tolerance against modified auto-antigens, including citrullinated proteins. This is followed by the appearance of serum autoantibodies in the absence of clinical manifestations. After a variable period of time, the individual at first may develop arthralgia without evidence of synovitis, which eventually evolves into a clinically manifest arthritis, classifiable at last as RA [4]. Mucosal surfaces play an important role in the immune response against foreign and potentially harmful environmental pathogens or toxins. Indeed, several risk factors for RA, such as cigarette smoking or dusts inhalation, act on lung parenchyma and may induce a subtle inflammatory state.

Lung inflammation is associated with an increased citrullination of tissue proteins and with local production of anti-citrullinated proteins antibodies (ACPA) not only in RA, but also in other chronic inflammatory conditions [5,6,7]. Sputum from RA patients as well as from subjects at risk for RA shows the presence of ACPA. Furthermore, ACPA can be found in sputum even before any evidence of them in serum [8]. Similarly, broncho-alveolar lavage from early, untreated RA is enriched in ACPA compared to serum [9]. This evidence is parallel to the demonstration of several subclinical lung abnormalities at high-resolution computed tomography (HRCT) in more than 70% of subjects positive for serum autoantibodies without arthritis [10].

The lung also represents a site of disease-related injury. RA-associated interstitial lung disease (RA-ILD) is a significant cause of morbidity and mortality among patients with RA [11]. The incidence of RA-ILD is around 10%, but it increases to 58% when subclinical lung abnormalities detected at HRCT are taken into account [12]. Although the clinical meaning of subclinical lung alteration is still unclear, there are reports of a potentially progressive nature up to the development of a clinically manifest RA-ILD [13,14]. Once developed, RA-ILD carries a poor prognosis; hence, early identification of those patients at risk for RA-ILD is of paramount importance, even if it is still unknown whether an immunosuppressive treatment can modify the course of the disease [15,16,17].

Pulmonary function testing (PFT) is commonly used as the first screening tool for RA-ILD identification. Impairment in pulmonary functional parameters, particularly reduction in diffusion lung capacity for carbon monoxide (DL_CO_), shows a good sensitivity in subclinical RA-ILD identification, although the clinical applicability of this finding remains unclear [17].

Cardiopulmonary exercise testing (CPET) provides a non-invasive, dynamic and global assessment of cardiopulmonary responses to physical exercise and represents a useful tool to assess the integrity of lung structures [18]. Its clinical utility has been increasingly recognized over the past decades [19]. In rheumatic conditions, CPET is commonly used in scleroderma-associated ILD, in which it has shown accuracy to identify early abnormalities and to provide several prognostic indications [20,21]. However, CPET utility in RA has not been extensively studied, especially regarding subclinical lung abnormalities.

Several serum biomarkers have been investigated for the identification of idiopathic pulmonary fibrosis (IPF). RA-ILD and IPF share several pathological and radiological features so that, predictably, validated IPF biomarkers showed good performance in RA-ILD identification as well. Among these, surfactant protein D (SPD) has been shown to be significantly increased in both clinically evident and subclinical RA-ILD [22]. SPD belongs to the collectin family and is primarily produced in type II pneumocytes and in Clara cells [23]. SPD plays an essential role in pulmonary innate immune defenses, enhancing pathogen clearance and regulating adaptive and innate immune-cell functions [24]. Circulating SPD has been correlated with a variety of pulmonary disorders [25], but has not been evaluated in the early phases of RA-associated lung abnormalities.

We hypothesized that, independently of the development of manifest RA, ACPA-positive subjects feature subclinical lung abnormalities, which could be non-invasively detected by pulmonary functional and SPD serum level assessment. Hence, the purposes of the present study were threefold:to evaluate the presence of subclinical lung involvement among ACPA-positive subjects without respiratory complains with and without confirmed diagnosis of RA;to assess pulmonary functional impairment at rest and during exercise through PFT and CPET, respectively;to evaluate the diagnostic potential of serum autoantibodies and SPD levels, with regard to radiology findings of lung involvement.

## 2. Results

### 2.1. Subjects’ Characteristics

A total cohort of 50 subjects was enrolled. At HRCT scan, two patients showed a “probable UIP” pattern and were consequently excluded from the study. The remaining 48 subjects, 37 female (77.1%) and 11 males (22.9%) were included in the study. The three study groups were composed by a total of 21, 10 and 17 ACPA-positive subjects without evidence of arthritis (ND), early RA (ERA) and long-standing RA (LSRA), respectively. Patients did not suffer from other significant comorbidities. The demographic, clinical and laboratory characteristics are summarized in Table 1. Within the LSRA group, eight patients (47.1%) were already on treatment with methotrexate (MTX), four patients (23.5%) were taking sulphasalazine and three patients (17.6%) hydroxychloroquine, either as monotherapy or as combination therapy. For the laboratory evaluation, 22 healthy control subjects (HC) were also enrolled, 15 females (68.2%) and 7 males (31.8%), the mean age being 47.3 ± 10.5 years.

### 2.2. Lung Function and Physiological Responses to Exercise

The results of the main PFT and CPET parameters are shown in Table 2.

Reduced DL_CO_ (i.e., <80% predicted value) was observed in 57.1%, 50% and 64.7% of ND, ERA and LSRA subjects, respectively (*p* = 0.747). There were no significant differences between groups in the examined PFT parameters, even after correction for smoking status. FEV_1_, when expressed as percentage of predicted value, was significantly lower in patients who were current or former smokers compared to non-smokers (*p* = 0.018). DL_CO_ values in current and former smokers were not different from those of never smoker subjects. 

All patients achieved maximal effort during CPET. Mean work rate at peak exercise was 105 ± 32.5 W (76.1 ± 15.4% predicted value). Reduced exercise tolerance, defined as V’O_2 peak_ < 80% predicted value, was found in 19 out of 48 (39.6%) patients. Even if not significantly different, ND and ERA subjects showed lower rates of exercise intolerance than LSRA. Anticipated θ_L_ (i.e., V’O_2_ at θ_L_ < 40% predicted V’O_2 peak_), indicating impaired aerobic fitness, was observed in six (12.5%) subjects, with no difference among subgroups. No abnormalities in cardiovascular responses were found, neither within the whole group nor among each subgroup.

Overall, there was no sign of ventilatory limitation, average V’E _peak_ being 49.8 ± 12% eMVV. In this regard, none of the recruited patients showed V’E _peak_ values > 85% eMVV. Mean V_T_ was 1.8 ± 0.5 L at peak exercise, corresponding to 50.0 ± 10.0% of FVC, confirming the absence of ventilatory constraints during exercise. Apparently, no differences in breathing patterns were observed between subgroups.

Peak SpO_2_ fell within normal values for all groups, although a significant trend in progressive reduction was found between groups (97.7 ± 1.1% vs 97.6 ± 1.2% vs 96.8 ± 1.7% among ND, ERA and LSRA patients, respectively; *p* = 0.008). Significant haemoglobin desaturation (peak—rest change in SpO_2_ > 4%) was found only in one patient with LSRA.

Impaired ventilatory efficiency, defined as V’E/V’CO_2_ at θ_L_ > 34 and/or V’E/V’CO_2_ slope > 30, was found in about one third of the study group. The highest frequency was registered among LSRA patients (41.2%); however, no significant differences were observed in the rate of reduced ventilatory efficiency among the three subgroups.

### 2.3. Laboratory Results

There were no significant differences in ACPA and RF levels between the groups. Significantly higher values of RF were registered among subjects from all the three groups with reduced exercise tolerance (263.8 ± 216.5 IU/L vs 119.4 ± 171.71 IU/ L; *p* = 0.019) and with impaired ventilatory efficiency (326.6 ± 282.9 IU/ L vs 90.8 ± 98.6 IU/ L; *p* = 0.015). No further relations were observed between ACPA and RF levels and pulmonary function at rest and during exercise.

SPD serum levels were significantly higher among study group subjects compared with HC (158.5 ± 132.3 ng/mL vs. 61.27 ± 34.11 ng/mL; *p* < 0.0001). Similarly, subgroup analysis revealed higher SPD levels within the ND (132.1 ± 125.1 ng/mL; *p* = 0.023), ERA (181.1 ± 151.3 ng/mL; *p* = 0.003) and LSRA subgroups (176.1 ± 131.8 ng/mL; *p* < 0.0001) compared with HC (61.2 ± 34.1 ng/mL) (Figure 1). No difference was observed between patient groups; nonetheless, a significant trend in increasing levels of SPD was noticed from ND to LSRA (*p* = 0.004). There were no differences in SPD levels based on smoking history (i.e., current, former or never smoker). However, significantly higher levels of SPD were present in ACPA-positive never smokers (ND+ERA+LSRA) compared with healthy controls (142.2 ± 94.1 ng/mL vs 61.2 ± 34.1 ng/mL; *p* < 0.0001). In the LSRA group, no difference was found in SPD serum levels concerning treatment with MTX. 

SPD levels were inversely related to V’O_2 peak_ (expressed as percentage of predicted value; *p* = 0.024; rho = −0.32), V’O_2_ at θ_L_ (percentage of predicted V’O_2 peak_; *p* = 0.013; rho = −0.36) and peak SpO2 (*p* = 0.008; rho = −0.38). As observed for RF, SPD levels were also shown to be significantly higher among patients with reduced exercise ventilatory efficiency (V’E/V’CO_2_ a θ_L_ > 34) (250.8 ± 155.3 ng/mL vs. 141.2 ± 121.9 ng/mL; *p* = 0.007). A similar result was also observed when limiting the analysis to ND patients (351.15 ± 219 ng/mL vs 93.5 ± 44.9 ng/mL; *p* = 0.01).

### 2.4. HRCT Abnormalities

Of the entire cohort, 62.5% had HRCT abnormalities. The most frequently detected abnormality was the presence of nodules, followed by evidence of fibrosis. The less frequent abnormality was the presence of air trapping. There were no differences in the frequency of the various HRCT abnormalities according to smoking status or, among LSRA patients, to treatment with methotrexate (MTX). The frequencies of HRCT abnormalities are shown in Table 3.

Subgroup analysis revealed significantly higher rates of overall HRCT abnormalities, nodules, emphysema and fibrosis among LSRA patients compared with the other subgroups (*p* = 0.001, *p* = 0.004, *p* = 0.02 and *p* = 0.003, respectively). The same differences in HRCT total abnormalities and nodules were also observed when limiting the analysis to those patients who never smoked (*p* = 0.049 and *p* = 0.016, respectively). Current and former smokers showed a significantly higher frequency of fibrosis compared with subjects who never smoked (*p* = 0.03), with a relative risk of 2.77 (CI 95% 1.054–8.359). Of note, no difference in fibrosis prevalence was found based on MTX treatment.

### 2.5. Relations between Functional and Laboratory Data and the Presence of HRCT Abnormalities

There were no differences in both dynamic and static lung volumes in patients with or without HRCT abnormalities. On the contrary, a significant reduction in DL_CO_ (74.19 ± 13.2% pred. vs. 131.7 ± 93% pred.; *p* = 0.009) and K_CO_ (77.5 ± 15.8% pred. vs 138.92 ± 97% pred.; *p* = 0.01) was found among patients with HRCT abnormalities compared with patients with normal HRCT scans. Impaired DL_CO_ was present only among patients with HRCT abnormalities who had already developed the disease (ERA and LSRA), but not in the ND group (*p* = 0.042). In the former group, the presence of a reduced DL_CO_ had a positive likelihood ratio for the presence of HRCT abnormalities (*LR* = 4.7). Reduction in DL_CO_ (73.8 ± 14.2% pred. vs 91.7 ± 18.5% pred.; *p* = 0.002) and K_CO_ (75.9 ± 16.6%pred. vs 96.1 ± 19% pred.; *p* = 0.003) was also present in patients with nodules at HRCT.

Patients with signs of ventilatory inefficiency at CPET also had an increased frequency of HRCT abnormalities. In particular, patients with increased V’E/V’CO_2_ at θ_L_ (i.e., >34) showed a higher frequency of overall HRCT abnormalities (*p* = 0.029) and specifically bronchiectasis (*p* = 0.009) and airways thickening (*p* = 0.035), while patients with an abnormal V’E/V’CO_2_ slope (i.e., >30) had an increased frequency of emphysema (*p* = 0.007). A significant difference in the rate of HRCT abnormalities according to V’E/V’CO_2_ at θ_L_ values persisted also when limiting the analysis to the ND group alone (*p* = 0.022).

SPD serum levels were significantly higher in subjects with HRCT abnormalities compared to subjects without abnormalities (213.5 ± 157.2 ng/mL vs 117.7 ± 157.3 ng/mL; *p* = 0.018) (Figure 2). Considering only patients who already had arthritis (ERA+LSRA groups), significantly higher levels of SPD were observed in subjects who had airways abnormalities (*p* = 0.023), but not in those who showed parenchymal abnormalities. There was a significant negative correlation in subjects with HRCT abnormalities between SPD serum levels and FVC % pred. (*p* = 0.039; rho = −0.465), FEV1/FVC % pred. (*p* = 0.005; rho = −0.599) and peak SpO_2_ (*p* = 0.033; rho = −0.477). In the same subjects, ACPA levels correlated with V’O_2_ at θ_L_ expressed both as absolute value (*p* < 0.001; rho = −0.726) and as percentage of predicted V’O_2 peak_ (*p* = 0.021; rho = −0.508).

In order to evaluate the diagnostic performance of SPD serum levels with regard to the detection of HRCT abnormalities, ROC analysis was performed (Figure 3). The AUC value was 0.77 (95% CI: 0.65–0.9). The optimal cut-off point of diagnostic performance was 90.78 ng/mL, with a sensitivity of 80% and a specificity of 62% for HRCT abnormalities detection.

Finally, in order to further evaluate the diagnostic potential of serum and functional biomarkers, a logistic regression model to find independent predictors of the presence of HRCT abnormalities was performed. This model found that an abnormal DL_CO_ (beta coefficient −2.9) and SPD (beta coefficient 0.009) stepped into the final model and retained statistically significant association with HRCT abnormalities, suggesting that normal DL_CO_ and increased SPD serum levels related to a decreased and increased likelihood of HRCT abnormalities (sensitivity 56.3%, specificity 86.4%), respectively (Table 4).

## 3. Discussion

The occurrence of subclinical lung abnormalities in the natural history of RA is frequent and largely dependent on the serological status. As mentioned above, Demoruelle et al. demonstrated the presence of subclinical lung abnormalities at HRCT in more than 70% of ACPA-positive subjects without evidence of arthritis [10]. The most frequent abnormalities were airways alterations, such as bronchial wall thickening and air trapping, with a minority of parenchymal alterations [10]. In a different cohort, Fisher et al. reported a prevalence of 54% of airways abnormalities, 14% of RA-ILD and 26% of a combination of both in ACPA-positive subjects without arthritis and with respiratory complaints [26]. Further evidence derives from the ancillary studies performed on the Multi-Ethnic Study of Atherosclerosis (MESA) cohort, a large, multi-centric cohort of healthy subjects undergoing CT scans for subclinical cardiovascular disease investigation. Within this cohort, the presence of high attenuation areas (HAA), a marker of subclinical ILD, was associated with the presence of RA-related autoantibodies [27]. Our results show that the transition from a systemic autoimmunity to the development of the disease is associated with a progressive increase in the prevalence of subclinical HRCT abnormalities, which are also frequently present before the onset of the arthritis. This increased frequency is already evident in the first years of the disease, considering that LSRA patients had a maximum disease duration of 3 years. This is in line with the finding of HRCT abnormalities in 68% of RA patients with a median disease duration shorter than 6 months described by Reynisdottir et al. [28]. Moreover, in up to one third of the cases, the diagnosis of symptomatic ILD is established between 1 year prior to and 1 year after the diagnosis of RA [29]. Globally, there is an increasing amount of evidence suggesting that the occurrence of HRCT abnormalities is an early event in the natural history of the disease, occurring in a parallel manner to joint involvement. In our study, the most frequently detected abnormalities were nodules. Although MTX treatment has been traditionally associated with accelerated nodulosis in RA [30], our study does not support the evidence of an increased prevalence of lung nodules in patients treated with MTX. Accordingly, a recent study on a large cohort of newly diagnosed RA confirmed that MTX treatment is not associated with the incidence of any kind of RA-ILD [31]. Conversely, our results confirm the association between smoking habit and pulmonary fibrosis risk in RA, as largely reported in the literature [32].

Several reasons may justify the importance of subclinical HRCT abnormalities detection in ACPA-positive subjects. Previous studies demonstrated a progressive nature of subclinical HRCT abnormalities, potentially evolving into clinical manifest ILD over a 2 year period [14]. Moreover, some evidence suggests that treatment of subclinical HRCT lung abnormalities that show a tendency to progress to ILD may stabilize the HRCT alterations [13]. The presence of subclinical lung abnormalities may also influence the decision regarding treatment options. In fact, there are several reports of new-onset ILD as well as worsening of pre-existing ILD for all available anti-TNF agents. However, the potential harm related to anti-TNF treatment is still not clear, with some studies reporting, on the contrary, a stabilization of lung function [33]. Indeed, specific manifestations of inflammatory lung involvement, such as bronchiectasis, may increase the risk of severe complications of biologic treatment [34]. To date, there are no specific recommendations about the diagnosis and treatment of RA-ILD, although a diagnostic algorithm has been proposed [35]. PFTs are informative in case of suspected RA-ILD. Previous studies reported a reduced DL_CO_ as a valuable marker of RA-ILD as well as of preclinical ILD in RA patients [14,36]. Our study confirms that subjects with subclinical HRCT abnormalities had subtle but statistically significant reductions in DL_CO_ compared with subjects without these alterations and that a reduced DL_CO_, expressed as percentage of predicted value, increased the likelihood of HRCT abnormalities. Notably, these associations were more evident in patients who already had developed manifest RA, not achieving statistical significance in the ND group.

Several studies investigated the diagnostic role of CPET in preclinical, non-rheumatologic conditions. In the MESA cohort, a ventilatory limitation to exercise was present in subjects with subclinical HAA at HRCT compared to subjects without HAA [37]. In a cohort of preclinical familial pulmonary fibrosis, CPET revealed that the percentage of reduction in dead space ventilation at peak exercise was significantly lower in subjects with asymptomatic ILD compared with subjects with normal HRCT scans [38,39]. We detected several CPET abnormalities in our patients, which could be associated with the early involvement of lung parenchyma. Indeed, the presence of a difference in SpO_2_ reached at peak exercise between groups may be related to the frequency of the subclinical HRCT abnormalities detected. In subjects with HRCT abnormalities, CPET showed a significant impairment in V’E/V’CO_2_ at θ_L_ and V’E/V’CO_2_ slope, suggesting a ventilation/perfusion mismatch. In our study, an abnormal V’E/V’CO_2_ relationship was associated with HRCT abnormalities involving both airways and lung parenchyma. This is in line with what has already been described across a wide spectrum of lung diseases, including emphysema and cystic fibrosis [40,41,42]. CPET may thus have a diagnostic value in RA patients, non-invasively suggesting the presence of HRCT abnormalities. Furthermore, the evidence of a similar association even among ND subjects suggests a possible utility of CPET for early identification of ACPA-positive subjects without arthritis who may be candidates for HRCT, for diagnostic or research purposes.

SPD is increasingly gaining attention for its potential role as a serum biomarker of RA-ILD. Within our study population, the fact that SPD was found to be significantly higher than healthy controls and the progressive increase in SPD serum levels from ND to LSRA groups is in agreement with the current knowledge of an early, subclinical lung involvement in the natural history of RA. Moreover, SPD may have a diagnostic role in HRCT abnormalities identifications, showing a good discriminative ability. Clara cells in small airways are one of the main sources of SPD [23]. The pulmonary expression of SPD increases to protect the lung against pathogens and to regulate the inflammatory response in the airways, as observed in chronic bronchitis exacerbations [43]. Accordingly, the negative correlation found between SPD and FVC % pred., FEV1/FVC % pred. and peak SpO_2_, as well as the finding of increased SPD serum levels among patients with airways involvement at HRCT, further suggests that SPD may be a selective marker of airway disease. Considering the difference in SPD serum levels between healthy controls and never smoker ACPA-positive subjects, the higher levels of SPD found in the latter may be directly related to the inflammatory process in the airways specifically associated with early phases of RA. Despite the presence of autoantibodies having been associated with an increased prevalence of subclinical ILD [27], we did not find any association between ACPA or RF levels and the presence of HRCT abnormalities. Anyway, all our patients were ACPA-positive, suggesting that the association with subclinical lung abnormalities depends on the serological positivity status for autoantibodies rather than the autoantibodies titer. RF levels showed a negative correlation with several CPET parameters of ventilatory efficiency, suggesting a higher lung involvement with increasing levels of autoantibodies. This observation may be related to a more significant functional pulmonary impairment with increasing levels of autoantibodies or, alternatively, may indicate the airways as a site of autoantibodies production as a consequence of harmful environmental stimuli [44,45].

This study presents several limitations. First, some of our results may be underpowered by the small sample size of this study. Larger cohorts could in fact reveal associations between the non-invasive markers and the various specific lung manifestations of the disease, which can be only supposed by the present study. Indeed, reliable markers of the most severe lung manifestations, such as fibrosis, may be relevant in clinical practice. A second limitation is represented by the cross-sectional nature of the study. The modification of the various functional and serum parameters during the evolution of the disease is currently unknown, limiting the prognostic potential of non-invasive assessment. Prospective cohorts are needed to address this issue, especially in the case of ACPA-positive subjects without arthritis. Finally, despite the observation of an abnormal ventilatory efficiency in 31.2% of the enrolled patients, a proper interpretation of ventilatory efficiency in our study group is limited by the absence of data concerning arterial CO_2_ partial pressure during exercise. This data is indeed essential in order to define how inappropriate the ventilatory response of the lung is to the amount of carbon dioxide produced during the effort [46]. Whether this phenomenon actually stems from uneven lung ventilation or from a reset of arterial CO_2_ set-point should be investigated with properly designed studies.

## 4. Materials and Methods

### 4.1. Study Subjects

ACPA-positive subjects referred to the Early Arthritis Clinic of Policlinico Umberto I Hospital/“Sapienza” University of Rome were consecutively enrolled. Participants were divided into three groups. The first group included ACPA-positive subjects without clinical evidence of arthritis (no disease subjects, ND). The second group included ACPA-positive Early Rheumatoid Arthritis (ERA) patients, diagnosed according to the 2010 American College of Rheumatology (ACR)/European League Against Rheumatism (EULAR) criteria, naïve to treatment and with a disease duration shorter than 6 months. The third group included established, ACPA-positive Rheumatoid Arthritis patients (long-standing Rheumatoid Arthritis, LSRA), with a disease duration shorter than 36 months, on treatment with disease-modifying antirheumatic drugs (DMARDs). Subjects of both sexes and aged between 18 and 65 years were included in the study. Exclusion criteria were: any persistent respiratory complains, personal history of any kind of lung disease (e.g., asthma, chronic obstructive pulmonary disease) chronic heart failure NHYA class II-III-IV, overlap autoimmune syndromes, ongoing treatment with glucocorticoids, presumed or established pregnancy. Finding of Usual Interstitial Pneumonia (UIP) patterns at HRCT was also considered an exclusion criterion (see below).

Each subject underwent clinical and clinimetric, laboratory, functional and imaging evaluation. Age- and sex-matched healthy controls (HC) were also enrolled for comparison concerning laboratory assessment. Written informed consent was obtained from all patients and the study was approved by the local Ethics Committee (Comitato Etico Policlinico Umberto I, Sapienza Università di Roma, study reference number 815/13).

### 4.2. Clinic and Clinimetric Evaluation

Detailed medical history was taken for each subject. Every ACPA-positive subject without evidence of arthritis was evaluated to assess the absence of current or previous arthritis. General and musculoskeletal physical examination was performed, and counts of involved joints were registered for all participants.

Clinimetric scales were administered at the time of enrolment to evaluate global disease activity (i.e., Visual Analog Scale, VAS) and to calculate the composite index of disease activity (i.e., Disease Activity Score 28, DAS28). Moreover, disease onset and laboratory data regarding erythrocyte sedimentation rate (ESR), C-reactive protein (CRP), rheumatoid factor (RF) and ACPA levels were collected for each subject.

### 4.3. Laboratory Evaluation

A 15 mL venous blood sample was collected from each enrolled patient and from HC subjects. Serum was separated from blood samples and stored at −20 °C. After samples were defrosted, SPD serum levels were evaluated through a commercial ELISA kit (Biovendor, Modrice, Czech Republic), following the manufacturer’s guidelines. Each sample from the same patient has been evaluated in duplicate, with a variation coefficient of 3.9%. The analytic limit for SPD detection was 0.01 ng/mL.

### 4.4. Functional Evaluation

Spirometry and nitrogen wash-out for measurement of dynamic and static lung volumes, respectively, as well as single-breath determination of DL_CO_, were performed for each patient through an automated lung function testing system (Quark PFT, Cosmed, Rome, Italy) according to the standards recommended by the American Thoracic Society (ATS)/European Respiratory Society (ERS) [47,48]. All measurements were recorded as raw value and percentage of predicted value [49,50].

On the same day of PFT, incremental symptom-limited CPET was performed for every participant on an electronically braked cycle ergometer through an automated testing system (OMNIA, Cosmed, Rome, Italy), in accordance with international recommendations [19,51]. A fixed work rate increment of 10 W min^−1^ was used. The test was continued until the point of symptom limitation (peak of exercise).

Oxygen uptake (V’O_2_), carbon dioxide output (V’CO_2_), minute ventilation (V’E), tidal volume (V_T_) and respiratory frequency (f_R_) were analysed breath-by-breath during the test. Heart rate (HR), ECG and haemoglobin saturation by pulse oximetry (SpO_2_) were continuously monitored whilst blood pressure was measured every two minutes from rest to peak of exercise. All measured and derived parameters [e.g., ventilatory equivalents for O_2_ and CO_2_ (V’E/V’O_2_ and V’E/V’CO_2_, respectively), end-tidal O_2_ and CO_2_ partial pressures (P_ET_O_2_ and P_ET_CO_2_, respectively)] were recorded and averaged every ten seconds. Lactate threshold (θ_L_) was non-invasively estimated by the use of the dual methods approach (V-slope and ventilatory equivalents methods) [52].

V’O_2_ at peak exercise (V’O_2 peak_) was normalized for body weight and expressed also as percentage of predicted value. Peak V’E response (V’E _peak_) was expressed as a raw value and relative to estimated maximal voluntary ventilation (eMVV), which was defined as forced expiratory volume in the 1st second (FEV_1_) × 40 [19].

Ventilatory efficiency was evaluated through the analysis of the relationship between V’E (*y* axis) and V’CO_2_. The linear phase of the V’E/V’CO_2_ relationship was detected on the V’E (*y* axis) on V’CO_2_ (*x* axis) plot, between the beginning of loaded exercise and the end of the isocapnic buffering period, which was identified when V’E/V’CO_2_ increased and P_ET_CO_2_ decreased. Linear regression was then applied and the V’E/V’CO_2_ slope and its intercept on the *y* axis were calculated. V’E/V’CO_2_ raw value at θ_L_ was also recorded.

The subject’s effort was considered maximal either if the respiratory exchange ratio reached ≥1.10 or if HR achieved ≥85% of maximal predicted value at peak exercise (f). CPET parameters were compared with the predicted normal values [53].

All PFT and CPET were executed and analyzed by two physicians blinded to patients’ clinical and laboratory features.

### 4.5. Imaging Evaluation

All ACPA-positive participants underwent HRCT of the chest by the use of a multidetector scanner (Somatom Definition Siemens, Erlangen, Germany) with helical supine inspiratory contiguous acquisition (5 mm); images were reconstructed at 0.6 mm every 20 mm with high-resolution algorithms. HRCT images were reviewed by 2 radiologists who were blinded to each subject with regard to disease status, in order to evaluate the presence of subclinical parenchymal abnormalities (i.e., emphysema, fibrosis, ground-glass opacities, consolidations, nodules) and/or airways abnormalities (bronchiectasis, airways thickening, air trapping) as described previously [10]. The occurrence of “definite UIP” or “probable UIP” pattern was considered an exclusion criterion [54].

### 4.6. Statistical Analysis

Continuous variables are shown as mean ± SD or as median (range) for normally and non-normally distributed data, respectively. Categorical variables are presented as frequencies. Comparisons of continuous variables between two groups were performed using an independent samples T test or Mann–Whitney U test, whilst comparisons between more than two groups were tested through the ANOVA (with Bonferroni’s correction for post hoc adjustment) or Kruskal–Wallis test, according to data distribution. Chi-squared analysis tested the differences between categorical variables. Logistic regression analysis was performed to assess the strength of association between HRCT abnormalities and clinical, laboratory and functional features of interest. The predictive capacity of SPD serum levels for the presence of HRCT abnormalities was analyzed using ROC curves. Cut-offs with sensitivity and specificity to discriminate subjects with HRCT abnormalities from subjects without them were calculated. The significance of the correlations was evaluated with Spearman’s rank correlation coefficient.

All statistical analyses were performed using the SPSS Statistics version 24.0 software package (SPSS Inc., Chicago, IL, USA), and a two-sided *p* value < 0.05 was considered statistically significant.

## 5. Conclusions

The role of the lung in RA is dual: a site of autoimmunity generation and of disease-related injury. The identification of subclinical lung abnormalities can be relevant in the management of the disease, but a reliable biomarker that can easily identify lung involvement in RA patients is still lacking in clinical practice. HRCT is the gold standard for the evaluation of lung parenchyma but entails a significant exposure to ionizing radiations. Stratifying the risk for lung abnormalities in the individual patients may help to identify who needs to be referred early to a HRCT scan.

This study shows that PFR and CPET may help to identify RA patients who have an increased likelihood of HRCT abnormalities, and provide information about the kind of lung involvement that may have an impact on the clinical management and the therapeutic decision. SPD shows a good accuracy in identifying HRCT abnormalities in ACPA-positive subjects, confirming its role as a biomarker of lung involvement. The combination of serum biomarkers and PFT may provide a safe and harmless tool to identify subjects who require further investigations. SPD and functional parameters, especially DL_CO_, also have a potential application among individuals in the preclinical phase of the disease, contributing to identifying ACPA-positive subjects who likely have subclinical lung abnormalities, for research and clinical purposes. However, larger studies are needed to validate the potential role of PFR, CPET and SPD in ACPA-positive subjects.

## Figures and Tables

**Figure 1 ijms-21-05162-f001:**
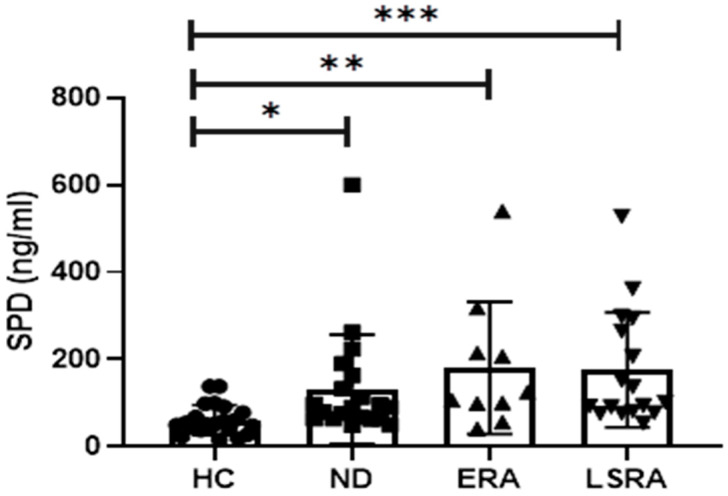
Comparison of SPD serum levels between ACPA-positive subjects and healthy controls. *: *p* < 0.05; **: *p* < 0.005; ***: *p* < 0.0005. Abbreviations: SPD, surfactant protein D; HC, healthy controls; ND, no disease subjects; ERA, early rheumatoid arthritis patients; LSRA, long standing rheumatoid arthritis patients.

**Figure 2 ijms-21-05162-f002:**
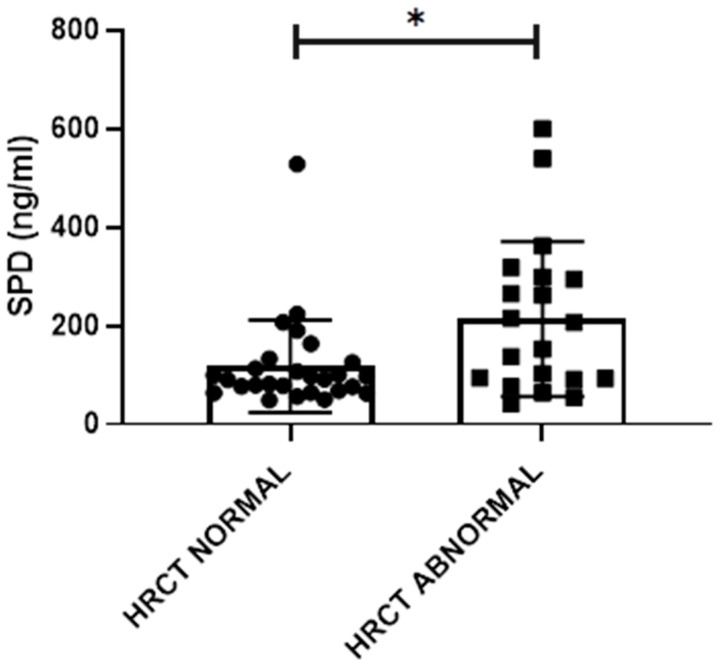
SPD serum levels in subjects with normal and abnormal HRCT. *: *p* < 0.05.

**Figure 3 ijms-21-05162-f003:**
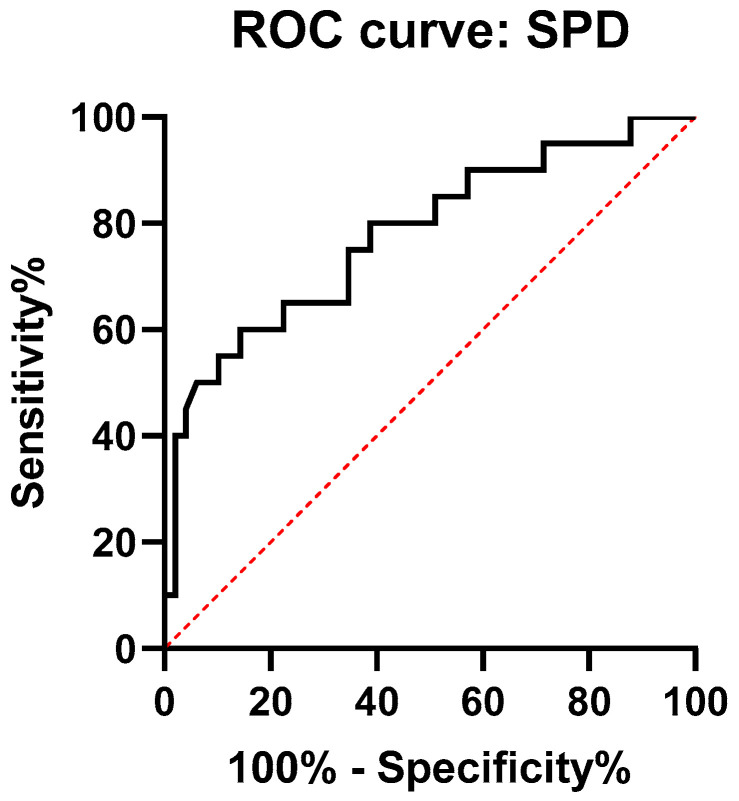
Receiver operating characteristic (ROC) curve of diagnostic performance of SPD serum levels in the identification of HRCT abnormalities.

**Table 1 ijms-21-05162-t001:** Demographic, clinical features and autoantibodies of enrolled subjects.

	Overall (*n* = 48)	ND (*n* = 21)	ERA (*n* = 10)	LSRA (*n* = 17)	HC (*n* = 22)	*p*
Age, years	49.8 ± 11	50.38 ± 13.6	48.5 ± 10.8	53.06 ± 7.3	47.3 ± 10.5	0.49
Sex, M/F	11/37	3/18	3/7	5/12	7/15	0.45
BMI, kg/m^2^	24.1 ± 4	23.3 ± 3.01	25.5 ± 5.6	25.5 ± 5.6	24.9 ± 2.8	0.28
Smoking, current/former/never	14/12/22	7/1/13	2/5/3	5/6/6	8/4/10	0.1
Disease duration, months	13.78 ± 11.02	-	3.6 ± 1.5	19.97 ± 9.6 ^†^	-	<0.0001
DAS28	3.6 ± 1.5	-	4.21 ± 1.7	3.2 ± 1.45	-	0.2
ACPA, UI/L	321.3 ± 394.9	342 ± 429.6	298.8 ± 202.2	309 ± 450.8	-	0.73
RF, UI/L	139.9 ±179.2	101.2 ± 138.8	182.2 ± 195.4	163 ± 212.5	-	0.46

Data are reported as mean ± SD, unless stated otherwise. *p* values intended for comparisons between ND, ERA, LSRA and HC (whenever applicable) subgroups of participants. ^†^: post hoc test *p* < 0.05 vs. ERA. Abbreviations: ND, no disease subjects; ERA, early rheumatoid arthritis patients; LSRA, long standing rheumatoid arthritis patients; HC, healthy controls; BMI, body mass index; ACPA, anti-citrullinated proteins antibodies; RF, rheumatoid factor.

**Table 2 ijms-21-05162-t002:** Selected pulmonary functional responses measured at rest and during exercise testing.

	Overall (*n* = 48)	ND (*n* = 21)	ERA (*n* = 10)	LSRA (*n* = 17)	*p*
FEV_1_, % pred.	102.6 ± 11.9	104.3 ± 11.0	100.5 ± 14.7	101.9 ± 11.5	0.56
FVC, % pred.	107.8 ± 12.8	108.3 ± 13.8	105.4 ± 9.3	108.7 ± 13.7	0.81
FEV_1_/FVC, %	80.2 ± 7.0	81.4 ± 6.9	80.0 ± 7.6	78.8 ± 6.9	0.7
TLC, % pred.	99.3 ± 12.1	98.5 ± 13.5	97.9 ± 11.9	101.3 ± 11.3	0.5
DL_CO_, % pred.	81.5 ± 16.6	83.7 ± 18.7	82.5 ± 19.4	78.5 ± 13.0	0.76
K_CO_, % pred.	84.3 ± 17.5	85.8 ± 17.8	90.2 ± 20.4	79.6 ± 15.3	0.21
Reduced DL_CO_, *n* (%)	28 (58.3)	12 (57.1)	5 (50.0)	11 (64.7)	0.71
Work rate _peak_, % pred.	76.1 ± 15.4	71.4 ± 14.2	71.1 ± 12.7	84.4 ± 15.4	0.35
V’O_2 peak_, mL/min/kg	22.7 ± 4.4	23.1 ± 3.9	23.6 ± 5.6	21.8 ± 4.3	0.81
V’O_2 peak_, % pred.	90.1 ± 15.9	92.6 ± 17.8	90.7 ± 14.5	86.7 ± 14.6	0.63
Reduced exercise tolerance, *n* (%)	19 (39.6)	7 (33.3)	3 (30.0)	9 (52.9)	0.53
V’O_2_ at θ_L_, % pred. V’O_2 peak_	52.7 ± 10.6	55.2 ± 9.6	47.3 ± 11.7	53.0 ± 10.6	0.2
V’E _peak_, l/min	57.3 ± 18.9	54.0 ± 10.1	65.0 ± 20.3	56.7 ± 24.9	0.39
V’E _peak_, %eMVV	49.8 ± 12.0	49.3 ± 11.8	54.5 ± 11.9	47.6 ± 12.2	0.63
SpO_2 peak_, %	97.3 ± 1.4	97.7 ± 1.1	97.6 ± 1.2	96.8 ± 1.7 *	0.017
ΔSpO_2_, %	-0.3 ± 1.3	-0.2 ± 1.2	0 ± 0.9	-0.7 ± 1.5	0.26
V’E/V’CO_2_ at θ_L_	30.6 ± 4.6	29.9 ± 5.0	31.6 ± 5.6	30.7 ± 4.6	0.72
V’E/V’CO_2_ slope	27.8 ± 4.6	27.0 ± 5.0	29.0 ± 3.8	28.0 ± 4.5	0.5
Impaired ventilatory efficiency, *n* (%)	15 (31.2)	6 (28.6)	2 (20.0)	7 (41.2)	0.5

Data are reported as mean ± SD. *P* values intended for comparisons between ND, ERA and LSRA subgroups of participants. *: post hoc test *p* < 0.05 vs. ND; Reduced DL_CO_: DL_CO_ < 80% of predicted value; reduced exercise tolerance: V’O2 peak < 80% of predicted value; impaired ventilatory efficiency: V’E/V’CO_2_ a θ_L_ > 34 and/or V’E/V’CO_2_ slope > 30. Abbreviations: ND, no disease subjects; ERA, early rheumatoid arthritis patients; LSRA, long standing rheumatoid arthritis patients; FEV_1_, forced expiratory volume; FVC, forced vital capacity; TLC, total lung capacity; DL_CO_, diffusing lung capacity for carbon monoxide; K_CO_, transfer coefficient of the lung; V’O_2_, oxygen uptake; V’CO_2_, carbon dioxide output; V’E, minute ventilation; θ_L_, lactate threshold; eMVV, estimated maximal voluntary ventilation; SpO_2_, peripheral capillary oxygen saturation; ΔSpO_2_, peak-rest change in peripheral capillary oxygen saturation.

**Table 3 ijms-21-05162-t003:** Frequencies of total and selected lung abnormalities at high-resolution computed tomography (HRCT) scan.

	Overall (*n* = 48)	ND (*n* = 21)	ERA (*n* = 10)	LSRA (*n* = 17)	*p*
Total, %	62.5	38	60	94.1 *^,†^	0.001
Parenchymal, %	62.5	38	60	94.1 *^,†^	0.001
Airways, %	16.6	14.3	10	23.5	0.98
Emphysema, %	16.6	-	30 *	29.4 *	0.02
Fibrosis, %	29.1	14.3	10	58.8 *^,†^	0.003
Ground glass, %	6.25	-	-	17.6	0.054
Consolidations, %	10.4	4.7	10	17.6	0.43
Nodules, %	50	28	60	76.4 *	0.004
Bronchiectasis, %	12.5	4.7	20	17.6	0.43
Airways thickening, %	16.6	14.3	10	23.5	0.61
Air trapping, %	8.3	-	10	11.8	0.14

Data are reported as percentage of the total. *P* values intended for comparisons between ND, ERA and LSRA subgroups of participants. *: post hoc test *p* < 0.05 vs. ND; ^†^: post hoc test *p* < 0.05 vs. ERA. Abbreviations: ND, no disease subjects; ERA, early rheumatoid arthritis patients; LSRA, long standing rheumatoid arthritis patients.

**Table 4 ijms-21-05162-t004:** Logistic regression model for predictors associated to the presence of HRCT abnormalities.

Dependent Variable	Predictors	B	SE	OR	95% CI	*p*	R^2^
HRCT Abnormal (Yes)	SPD	0.009	0.05	1.009	1.000–1.018	0.041	
	Reduced DL_CO_ (No)	−2.94	1.26	0.053	0.04–0.63	0.02	
	ACPA	0.001	0.001	1.001	0.998–1.003	0.44	
	V’E/V’CO_2_ at θ_L_ Impaired (No)	−0.65	1.32	0.52	0.039–6.9	0.62	
	Constant	−0.629	1.477	0.53		0.67	
							0.506

Variables included in the model: SPD, reduced DL_CO_ (i.e., <80% of predicted value), ACPA, impaired V’E/V’CO_2_ at θ_L_. (i.e., >34). Abbreviations: SPD, surfactant protein D; DL_CO_, diffusing capacity for carbon monoxide; ACPA, anti-citrullinated proteins antibodies.

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
