# Peer review of "Identification of Subclinical Lung Involvement in ACPA-Positive Subjects through Functional Assessment and Serum Biomarkers"

_ijms, 2020, doi:10.3390/ijms21145162_

Round 1
Reviewer 1 Report
This is an interesting paper on the early detection of lung involvement in ACPA positive patients.
I have a few questions and comments:
I think one of the biggest limitations of the study is the small sample size and the lack of validation of these results in a different cohort.
Patients characteristics – what were the comorbidities of the patients? SP-D is a molecule that has been studied and found to be elevated in several diseases (pulmonary, kidney disease, etc) and so the presence of other diseases could be alter the levels of SP-D.
I would like to see a comparison of the model that includes PFTs, SP-D and CPET for the prediction of lung involvement with the model that only uses serum concentrations of SP-D. Is the combined model superior?
Line 116 All the patients were receiving only monotherapy? Would you say RA in your patients was not so severe?
Line 145 exclude the word ‘performed’
Line 148 – authors find that non diseased subjects and early arthritis patients have higher frequency of exercise intolerance that long-standing arthritis patients? Is this correct? How is this explained? They mention no cardiac function abnormalities were found in any patients.
Line 167 – this correlation is quite weak – and only 6 patients had impaired aerobic fitness.
Line 168 – which patients with reduced exercise tolerance, all 3 groups or the ones in a specific group? In line 148 the authors mention that ND and ERA patients had lower exercise tolerance than the LSRA.
Line 169 – is this seen in all patients with reduced ventilatory efficiency? Was there difference by group, since the authors state that a third of all patients had reduced ventilatory efficiency but no difference was observed between the groups?
Line 176 - based on the current, former….
Line 179 - substitute “regarding”, maybe with “in relation to” or whatever the authors consider adequate
Line 187 – Similar to line 167 – were there differences between the groups? a third of all patients had reduced ventilatory efficiency but no difference was observed between the groups.
Line 190 - Of the entire cohort
Line 240 – What is the specificity and sensitivity of the combined model? The authors propose a combination of SPD, CPET and PFTs should be used for detection of lung involvement in ACPA positive patients. Is this model better than just the use of the serum SP-D? If not, why do the authors recommend this combination? It has higher costs and strong arguments are required to recommend this combination for the screening of ACPA positive patients. I am not sure how feasible it is in clinical practice.
Line 253 – dependent ON the serological
Line 254 – abnormalities ON HCRT
Line 257 – 259 Fisher et al. reported a prevalence of 54% of airways abnormalities, 14% of RA-ILD and 26% of a combination of both in ACPA-positive subjects without arthritis and with
respiratory complains.
Line 282 “worsening” – not adequate word
Line 288 – on the contrary
Line 289 – complications
Line 291 – PFTs
Line 324 – is in agreement instead of agreement
Line 332 – further suggests
Line 334 – found instead of founded
Line 338 – depends on
Line 344 – harmful
Line 347 – referred
Line 360 – on HCRT
Line 367 – enrollment
Line 376 – has been evaluated, “with” a variation coefficient instead of “using”
Line 415 – underwent to high
Line 417 – inspiratory contiguous acquisition
Line 451 – harmless
Line 452 – also have a potential
Author Response
Dear Reviewer,
Many thanks for your precious suggestions to improve the paper. Please, find the answers to your points as follow:
I think one of the biggest limitations of the study is the small sample size and the lack of validation of these results in a different cohort.
We definitely agree with your opinion, the sample size represents the major limit. We provided a paragraph in the discussion section to clearly point out the study limits (lines 341-356). From another point of view, the main strength of the study is to systematically investigate the lung involvement using several different non-invasive approaches, which has not been reported in literature.
Patients characteristics – what were the comorbidities of the patients? SP-D is a molecule that has been studied and found to be elevated in several diseases (pulmonary, kidney disease, etc) and so the presence of other diseases could be alter the levels of SP-D.
The main comorbidities which may represent bias for SP-D serum levels, CPET and PFT (lung diseases, cardiovascular disease) have been included into the exclusion criteria. No other significant comorbidities associated to increased SP-D levels were present among patients. We added a sentence in line 111
I would like to see a comparison of the model that includes PFTs, SP-D and CPET for the prediction of lung involvement with the model that only uses serum concentrations of SP-D. Is the combined model superior?
As you suggested, we evaluated different models in the prediction of the presence of HRCT abnormalities. A model including ACPA, SP-D, DLCO abnormalities and VE/VCO2 at LT abnormalities, although retaining as significant predictors only SP-D and DLCO, showed a better explanation of the variance (R square 0,506) respect the originally presented model. Accordingly, we decided to modify the originally presented model. Please, find here the other requested model to compare:
Originally presented model (FIGURES CAN BE FOUND IN PDF FILE ATTACHED)
ACPA and SP-D model
Only SP-D model
Line 116 All the patients were receiving only monotherapy? Would you say RA in your patients was not so severe?
Patients were receiving either monotherapy than combination therapy, we provided to specify in line 114-115. According to DAS28, patients showed a high disease activity in ERA group (they were early and untreated) and a moderate mean disease activity in LSRA group (as result of the treatment).
Line 145 exclude the word ‘performed’
Modified
Line 148 – authors find that non diseased subjects and early arthritis patients have higher frequency of exercise intolerance that long-standing arthritis patients? Is this correct? How is this explained? They mention no cardiac function abnormalities were found in any patients.
Thank you, this was a mistake. As you can see in the table, ND and ERA showed a lower exercise intolerance. We provided the correction. We can only speculate that LSRA patients, for the articular and the extra-articular more severe involvement, may have higher limitation in the exercise, but considering that the data is not statistically significant, we decided to not further discuss it.
Line 167 – this correlation is quite weak – and only 6 patients had impaired aerobic fitness.
We agree with your comment. We removed the sentence from the text.
Line 168 – which patients with reduced exercise tolerance, all 3 groups or the ones in a specific group? In line 148 the authors mention that ND and ERA patients had lower exercise tolerance than the LSRA.
In this case, we refer to all patients with a reduced exercise tolerance. No differences were present among the single groups. We already provided the correction in line 148, actually ND and ERA patients showed higher exercise tolerance. We modify the text to specify that the observation was on the overall cohort.
Line 169 – is this seen in all patients with reduced ventilatory efficiency? Was there difference by group, since the authors state that a third of all patients had reduced ventilatory efficiency but no difference was observed between the groups?
Again, also in this case, we refer to the overall cohort. We provided a better specification in the text
Line 176 - based on the current, former….
Modified
Line 179 - substitute “regarding”, maybe with “in relation to” or whatever the authors consider adequate
Modified as you suggested
Line 187 – Similar to line 167 – were there differences between the groups? a third of all patients had reduced ventilatory efficiency but no difference was observed between the groups.
A significant higher values of SP-D was present also in ND subjects with an abnormal V'E/V'CO2 at the LT compared with subjects with a normality of this parameter. We provided it in the text
Line 190 - Of the entire cohort
Modified
Line 240 – What is the specificity and sensitivity of the combined model? The authors propose a combination of SPD, CPET and PFTs should be used for detection of lung involvement in ACPA positive patients. Is this model better than just the use of the serum SP-D? If not, why do the authors recommend this combination? It has higher costs and strong arguments are required to recommend this combination for the screening of ACPA positive patients. I am not sure how feasible it is in clinical practice.
The sensitivity and the specificity of the combined model are respectively 56.3 % and 86.4% (we reported it in the main text), with a percentage accuracy in classification of 73.7 %, while in the model including only SP-D the sensitivity and the specificity are respectively 50% and 88.9 %, with a percentage accuracy in classification of 72.3%. The accuracy of the 2 models are consequently really similar. Considering that, a part from SP-D, only the abnormal DLCO is a significant predictor in the model, we modified the statement in the “Conclusion” section about the utility of the combination tests in the subclinical lung identification. Although we strongly agree with your perplexity of the cost-effectiveness of the CPET for this purpose, we are aware that the majority of RA patients undergoes to PFT even at the beginning of the disease management (just consider the habit of many rheumatologists to screen lung function before MTX prescription). As consequence, DLCO data are often available in RA patients, so we think this data about PFT would not lead to an excessive resort to PFT for this specific purpose. By contrary, even the slight increase in sensitivity could save from a variable number of HRCT scans, way more expensive. Of course, we agree with your point of the need of strong argument to recommend CPET in this specific setting.
Line 253 – dependent ON the serological
Modified
Line 254 – abnormalities ON HCRT
Modified
Line 257 – 259 Fisher et al. reported a prevalence of 54% of airways abnormalities, 14% of RA-ILD and 26% of a combination of both in ACPA-positive subjects without arthritis and with respiratory complains.
Modified as you suggested
Line 282 “worsening” – not adequate word
Modified with evolving
Line 288 – on the contrary
Modified
Line 289 – complications
Modified
Line 291 – PFTs
Modified
Line 324 – is in agreement instead of agreement
Modified
Line 332 – further suggests
Modified
Line 334 – found instead of founded
Modified
Line 338 – depends on
Modified
Line 344 – harmful
Modified
Line 347 – referred
Modified
Line 360 – on HCRT
Modified
Line 367 – enrollment
Modified
Line 376 – has been evaluated, “with” a variation coefficient instead of “using”
Modified
Line 415 – underwent to high
Modified
Line 417 – inspiratory contiguous acquisition
Modified
Line 451 – harmless
Modified
Line 452 – also have a potential
Modified
Thank you again for your contribution
All the best,
The Authors

Reviewer 2 Report
Dear authors,
thank you for this study on lung involvement in ACPA+ subjects. Although this paper shows some trends I think the authors need to acknowledge some limitations of their research:
- One of the issues here is the very small sample size and the multitude of theories tested. I personally would add a limitation section to discussio and critically review how low sample size could have influenced the results
- Of course, I will give some clues. Firstly, multiple testing can be an issue. I personally would think a Bonferonni correction is too strict but 0.05 yields a very high chance for false positives. I would suggest 0.01 or lower. One can see this in the log regression. A p=0.038 for SPD is indeed significant but this easily can be caused by an outlier.
- Please describe selection of patients AND healthy controls in methods. I was surprised discovering healthy controls after some time in the results. I would also give demographics of the healthy controls, to see if a comparison is truly just between controls and subjects.
- I wonder the effect of treatment of the patients on the lung abnormalities. Was this taken into account? As authors mention, treatment and RA-ILD are still unraveled.
- There are some issues with formatting here. In English its for example 0.0001 and not 0,0001 (a point not a comma). Anthropometric is a cool word but demographics is more used. P-values should be given instead of just stating "ns"
- For future research, it would also be nice to have a ACPA- cohort to control for.
- In abstract, i would delete "strictly" in the first sentence.
Good luck!
Author Response
Dear Reviewer,
Many thanks for your precious suggestions to improve the paper. Please, find the answers to your points as follow:
One of the issues here is the very small sample size and the multitude of theories tested. I personally would add a limitation section to discussio and critically review how low sample size could have influenced the results
We definitely agree with your point. We provided a critical review of the limits of the study in the “Discussion” section (lines 341-356).
Of course, I will give some clues. Firstly, multiple testing can be an issue. I personally would think a Bonferonni correction is too strict but 0.05 yields a very high chance for false positives. I would suggest 0.01 or lower. One can see this in the log regression. A p=0.038 for SPD is indeed significant but this easily can be caused by an outlier.
Being conscious of the small sample size, all the multiple testing have been performed using the Bonferroni correction, and consequently all the significances reported in the text are adjusted. We reported that all the significances are adjusted for Bonferroni in the methods section. For the logistic regression, we performed a new analysis trimming out the outliers, here reported
(FIGURES AVAILABLE IN THE PDF FILE)
As you can see, the significance of the predictors and the performance of the model are maintained also with this correction. These two points strengthen our confidence in the solidity of the reported data.
Please describe selection of patients AND healthy controls in methods. I was surprised discovering healthy controls after some time in the results. I would also give demographics of the healthy controls, to see if a comparison is truly just between controls and subjects.
We provided both in methods and in the results sections the requested data (lines 115-116, 373-374)
I wonder the effect of treatment of the patients on the lung abnormalities. Was this taken into account? As authors mention, treatment and RA-ILD are still unraveled.
This is a really interesting point. As stated in the “HRCT abnormalities” subsection, we did not find any difference in HRCT abnormalities based on the various treatments, especially MTX. However, the study was not designed to evaluate the influence of the treatment on lung abnormalities, especially for its cross-sectional design. The need for a prospective study on this topic has been pointed out in the new limit section.
There are some issues with formatting here. In English its for example 0.0001 and not 0,0001 (a point not a comma). Anthropometric is a cool word but demographics is more used. P-values should be given instead of just stating "ns"
We provided the requested changes. We also provided the non significant p-values
For future research, it would also be nice to have a ACPA- cohort to control for.
We definitely agree with this. Given the importance of lung pathology in ACPA generation, the confrontation versus seronegative RA may provide several pathogenic insights.
In abstract, i would delete "strictly" in the first sentence.
We provided the requested change.
Thank you again for your contribute,
All the best
The Authors

Round 2
Reviewer 2 Report
Dear authors,
the manuscript is substantially improved.
However,
- I need more info about the demographics of the HC in table1.
- in abstract, i would note the main outcome with numbers and p-values
- text needs serious language editing. New limitation section especially.
Good luck
Author Response
Dear Reviewer,
thanks again for your suggestions.
We provided the demographics of HC in table 1 and the main outcomes in the abstract. We performed also an extensive spell and grammar check of the whole text.
All the best
The Authors
Round 3
Reviewer 2 Report
Dear authors,
Thank you for these substantial improvements to the text. I have no further comments.